# Multifunctional Oxidized Dextran as a Matrix for Stabilization of Octahedral Molybdenum and Tungsten Iodide Clusters in Aqueous Media

**DOI:** 10.3390/ijms241210010

**Published:** 2023-06-11

**Authors:** Ekaterina V. Pronina, Yuri A. Vorotnikov, Tatiana N. Pozmogova, Alphiya R. Tsygankova, Kaplan Kirakci, Kamil Lang, Michael A. Shestopalov

**Affiliations:** 1Nikolaev Institute of Inorganic Chemistry SB RAS, 3 Acad. Lavrentiev Ave., Novosibirsk 630090, Russia; pronina@niic.nsc.ru (E.V.P.); tnp_post@mail.ru (T.N.P.); alphiya@niic.nsc.ru (A.R.T.); shtopy@niic.nsc.ru (M.A.S.); 2Institute of Inorganic Chemistry of the Czech Academy of Sciences, Řež 1001, 250 68 Husinec-Řež, Czech Republic; kaplan@iic.cas.cz (K.K.); lang@iic.cas.cz (K.L.)

**Keywords:** octahedral iodide cluster, molybdenum, tungsten, oxidized dextran polysaccharide, hydrolysis, stability, luminescence, cytotoxicity, photodynamic therapy

## Abstract

Due to their high abundance, polymeric character, and chemical tunability, polysaccharides are perfect candidates for the stabilization of photoactive nanoscale objects, which are of great interest in modern science but can be unstable in aqueous media. In this work, we have demonstrated the relevance of oxidized dextran polysaccharide, obtained via a simple reaction with H_2_O_2_, towards the stabilization of photoactive octahedral molybdenum and tungsten iodide cluster complexes [M_6_I_8_}(DMSO)_6_](NO_3_)_4_ in aqueous and culture media. The cluster-containing materials were obtained by co-precipitation of the starting reagents in DMSO solution. According to the data obtained, the amount and ratio of functional carbonyl and carboxylic groups as well as the molecular weight of oxidized dextran strongly affect the extent of stabilization, i.e., high loading of aldehyde groups and high molecular weight increase the stability, while acidic groups have some negative impact on the stability. The most stable material based on the tungsten cluster complex exhibited low dark and moderate photoinduced cytotoxicity, which together with high cellular uptake makes these polymers promising for the fields of bioimaging and PDT.

## 1. Introduction

Polysaccharides are very abundant polymeric carbohydrates, most of which are of natural origin. These polymers are composed of simple monosaccharide units linked together via glycoside bonds. In addition to their relatively low cost resulting from their production from renewable sources, polysaccharides are non-toxic, biodegradable, and biocompatible compounds [1], which makes them one of the most important representatives of biopolymers and explains the great interest of researchers in various fields. Polymeric structure in combination with a large number of modifiable hydroxyl groups (-OH) determined one of the most important applications of polysaccharides—stabilization of various compounds or nanoparticles in aqueous media [2,3,4,5]. Indeed, they have proven to be efficient stabilizing agents for gold nanoparticles [6,7,8], quantum dots [9,10,11,12,13,14], upconversion nanoparticles [15,16,17], etc. Hybrid materials based on them can be used in various fields of biology and medicine, e.g., as biosensors [8,11,12,13], agents for bioimaging [9,10,11], and photothermal [17] and photodynamic therapies [14,15,16].

Dextran, a branched glucose polymer widely used in practical medicine, occupies a special place among a large number of polysaccharides [18]. In the main chain of this polysaccharide, the monomeric D-glucopyranose fragments are predominantly linked by α-1,6-glycosidic bonds, and the side branches are α-1,3 bonds. Previously, our group developed a simple and eco-friendly method for modifying dextran by oxidation with hydrogen peroxide at elevated temperatures [19]. During this process, carbonyl (-HC=O), acidic (-COOH), and peroxo (-C(O)OOH) groups are formed. The peroxo groups can be easily removed in an alkaline medium to produce an ionic, non-toxic polymer. The resulting multifunctional polymer is an excellent candidate for the stabilization of nanoscale objects.

Octahedral molybdenum and tungsten halide cluster complexes of the general formula [{M_6_X_8_}L_6_]^n^ (M = Mo or W, X = halogen, L = terminal ligand) have great potential for biomedical applications thanks to their bright luminescence with high photophysical parameters and ability to generate singlet oxygen [20,21,22,23,24,25,26]. Unfortunately, the insolubility of the majority of known clusters in water as well as the low hydrolytic stability of a modest number of reported water-soluble complexes [27,28,29,30] hinder their application in living systems in molecular form. Incorporation of cluster complexes into matrices of different natures has proven to be an efficient approach to stabilizing clusters in aqueous solutions [21,23,25,26,31,32,33,34,35], but utilization of water-soluble polymers has not been widely developed.

Here we report the stabilization of water-soluble molybdenum and tungsten cluster complexes [{M_6_I_8_}(DMSO)_6_](NO_3_)_4_ (M = Mo, W) [29], previously obtained in our group, in aqueous solution and in culture medium, via incorporation into a multifunctional oxidized dextran containing aldehyde and carboxyl groups. The effect of the functional group ratio and molecular weight of the modified dextran on its ability to stabilize cluster complexes was studied. For the most stable sample, biological studies were performed on Hep-2 cells (human larynx carcinoma cell line); namely, dark and photoinduced cytotoxicity were studied by MTT assay, while cellular uptake kinetics were traced by flow cytometry.

## 2. Results and Discussion

### 2.1. Preparation and Characterization of M^x^@DexQ and M^x^@NaOxDexQ(n)

In this work, the natural, biocompatible, water-soluble polymer dextran and its oxidized derivatives were chosen as a stabilizing matrix. We have previously demonstrated [19] that hydroxyl groups in polysaccharides can be oxidized by hydrogen peroxide to form carbonyl, carboxylic, and peroxo groups. The following treatment of an aqueous solution of oxidized dextrans with sodium hydroxide to a neutral pH leads to the destruction of peroxo groups and the formation of nontoxic sodium salts of oxidized polymers. The content of functional groups and molecular weight of the final polymer depend on the amount of oxidizing agent as well as the molecular weight of the initial dextran (Appendix A). Here we used dextrans with molecular weights (Q) of 6 (Dex6) and 60 (Dex60) kDa and sodium salts of the corresponding oxidized dextrans—NaOxDexQ(n), where n = 1 (polymer oxidized using a 2.0 M solution of H_2_O_2_) and 2 (polymer oxidized using a 5.0 M solution of H_2_O_2_). These polymers were chosen due to their highest diversity in terms of the content of the functional groups and the molecular masses (Appendix A), which should allow us to identify the parameters affecting the stabilization of the cluster complexes. As a target of stabilization, water-soluble complexes [{M_6_I_8_}(DMSO)_6_](NO_3_)_4_ were chosen since their hydrolytic stability was studied in detail in our earlier works [29,30].

To obtain the materials M^x^@NaOxDexQ(n) (where M is Mo or W; x = 100 for DexQ and x = 10, 50, and 100 for NaOxDexQ(n) is the loading of cluster complex [{M_6_I_8_}(DMSO)_6_](NO_3_)_4_ in mg per 100 mg of polymer), a certain amount of cluster complex and dextran were dissolved in DMSO, and the resulting solution was stirred for 24 h at room temperature. Orange (for M = Mo) or yellow (for M = W) cluster-containing materials were precipitated by the addition of the excess of ethanol, while the unbound complex remained in solution. To eliminate weakly bonded unwashed complexes, the polymer was dissolved in water and reprecipitated with ethanol. We assume that during the synthesis, aldehyde (carbonyls) and/or acid (carboxylates) groups of functionalized dextran can be covalently bonded to the cluster core, or terminal H_2_O/OH^−^ ligands of partially hydrolyzed clusters can form hydrogen bonds with any of the groups of the polymer (Figure 1).

FTIR spectroscopy is a useful tool to monitor bonding modes realized in materials according to characteristic shifts of vibration bands of a certain group. In the case of materials based on non-oxidized dextrans, no noticeable differences in the FTIR spectra of DexQ/M^x^@DexQ pairs were observed. The FTIR spectra of all the materials based on oxidized polymers also correspond in general to pure oxidized dextrans and its sodium salts (Figure 2A, Appendix A), but some important changes are observed in the range of functional group oscillations (~1550–1850 cm^−1^) (Figure 2B). After the oxidation of the initial polymer with hydrogen peroxide, a strong vibration band of CO groups belonging to carbonyls and carboxylates appears at ~1735 cm^−1^ (Figure 2B, purple line). When sodium salt of dextran is formed, the oscillations of COO^−^ groups shift to lower wavenumbers (~1605 cm^−1^) (Figure 2B, cyan line), resulting in a decrease in the intensity of the band at ~1735 cm^−1^. In turn, incorporation of the clusters in the polymer does not affect the band of carbonyl groups at ~1735 cm^−1^ but shifts the oscillation of COO^−^ groups to higher wavenumbers (~1620 cm^−1^) (Figure 2B, orange line), indicating coordination of carboxylates to metal centers in the cluster core. Indeed, it was shown that coordination of carboxylic acids with clusters results in similar shifts in comparison with corresponding salts [36]. No other signals related to cluster complexes or coordinated DMSO molecules were observed.

The content of molybdenum and tungsten in the obtained materials was determined using inductively coupled plasma atomic emission spectroscopy (ICP-AES) (Appendix A). Taking into account the preservation of cluster core, metal content was recalculated to the content of {M_6_I_8_} in μmole per gram of material, and its dependence on cluster loading (x) was plotted (Figure 3). Furthermore, theoretical values of {M_6_I_8_} content at 100% inclusion of the complex in the polymer were calculated and added to the graph (gray dashed lines).

Concerning non-oxidized dextrans, polymers with a higher molecular weight (Dex60 vs. Dex6) are able to incorporate a higher amount of cluster complexes for both M = Mo and W, indicating some impact of M_w_ on the overall capacity of the polymer. As expected, dextran oxidation results in an increase in cluster content. In the case of Mo^x^@NaOxDexQ (n), for x = 10, the real content of molybdenum is close to the theoretical one, while for x = 50 and 100, the values are lower than calculated and tend to reach a plateau (Figure 3A). The difference in impregnation degree of the cluster in the case of different polymers (NaOxDex6(n) vs. NaOxDex60(n)) is negligible, indicating that functional group loading and molecular weight of oxidized dextrans do not significantly affect the capacity of the matrix. Nevertheless, some tendency toward higher inclusion in more oxidized polymers was observed. In turn, regarding tungsten cluster incorporation into oxidized dextrans, the metal content increases linearly with an increase in the loading of the cluster complex (x) (Figure 3B). Similar to Mo^x^@NaOxDexQ(n), up to x = 50, no effect of functional group loading or molecular weight of dextrans on the incorporation of tungsten clusters in polymers was observed. At x = 100, NaOxDex60(n) contains ~1.5 times more complex than NaOxDex6(n), regardless of oxidation degree (n). Considering the close molecular weights of NaOxDex6(n) and NaOxDex60(2) (Appendix A), M_w_ does not significantly affect matrix capacity. In terms of functional groups, at similar n, NaOxDex60(n) contains ~2 times more carbonyl groups and ~1.5 times fewer carboxyl groups than NaOxDex6(n) (Appendix A). This can indicate either that carbonyl groups play an important role in cluster binding or that a high number of acidic groups prevents efficient incorporation of the cluster. In general, at x = 100, the oxidized dextran NaOxDex60(n) contains a comparable amount of molybdenum and tungsten clusters.

Furthermore, the sulfur content of the obtained materials was determined using ICP-AES. In general, there are 3–4 DMSO molecules per cluster core {M_6_I_8_}, indicating that partial hydrolysis and/or binding with functional groups of dextrans (carboxyl or carbonyl) occurs during the synthesis.

### 2.2. Stability of the Materials in Water and Culture Medium

According to our earlier studies, initial cluster complexes [{M_6_I_8_}(DMSO)_6_](NO_3_)_4_ (M = Mo, W) are gradually hydrolyzed in pure water [29]. To evaluate the efficiency of dextrans as stabilizing agents and the impact of different factors (such as type and number of functional groups as well as molecular weight of the polymer) on the overall stability, the changes in UV-vis spectra of aqueous solutions of the materials obtained with time were monitored (Appendix A). Interestingly, the shape of the spectra of the fresh materials is close but not similar to the spectra of corresponding pure cluster complexes (Appendix A), indicating partial hydrolysis of the clusters in the polymer during the synthesis, which agrees with FTIR and ICP-AES data. In the case of materials based on the initial polymers (M^x^@DexQ), a gradual decrease in absorption with the complete disappearance of cluster-related bands after ~3 days of aging was observed. All the materials based on oxidized dextrans demonstrated exceptional stability with minor changes in the absorption intensity or the profile shape up to 13 days (Appendix A). According to ICP-AES (Mo and S content), aged for two weeks, Mo^100^@NaOxDex60(1), obtained by precipitation with EtOH from the aqueous solution, does not contain any sulfur, indicating complete replacement of terminal DMSO ligands, probably with H_2_O, with the formation of water-soluble forms of aqua-hydroxo complexes [29,31]. Most likely, these forms are efficiently stabilized with dextran via the formation of hydrogen bonds with functional groups or their direct coordination with the cluster through covalent bonds. Such high stability of M^x^@NaOxDexQ(n) samples in an aqueous solution prevents the investigation of different parameters affecting the stabilization efficiency of dextrans.

Consequently, it was decided to study the stability of the materials in conditions close to physiological, namely, in DMEM culture medium (pH = 7.4, high content of inorganic salts, amino acids, polysaccharides, and proteins; see complete composition of DMEM in ESI, Appendix A). The addition of aqueous solutions of the initial complexes [{M_6_I_8_}(DMSO)_6_](NO_3_)_4_ in culture medium leads to quick precipitation (complete discoloration of the solution after ~3 h). According to elemental analyses by EDS and CHNS, the precipitates containing M and I in the ratio of 6:8 and not containing sulfur, phosphorus, chlorine, carbon, or nitrogen are most likely neutral aqua-hydroxo (AH) complexes, which is in agreement with our previous studies on pure [{M_6_I_8_}(DMSO)_6_](NO_3_)_4_ [29,30]. In the case of the samples based on non-oxidized dextrans, a similar trend was observed, indicating the low stabilizing efficiency of pure polymers in such harsh conditions. At the same time, oxidized dextrans significantly increased the stability of both clusters in the culture medium. For the samples based on oxidized polymers, the changes in UV-vis spectra with time were monitored (Figure 4, Appendix A). Overall, the process of the materials hydrolysis can be divided into two stages: (i) substitution of remaining DMSO ligands with water molecules or functional groups of the polymers, which agrees with our previous observations for the materials in pure water; and (ii) pH-induced deprotonation of H_2_O ligands or direct coordination of OH^−^ resulting in the precipitation of insoluble AH complexes. At the first stage, only changes in the form of the spectra profile occur. The beginning of the second stage leads to a gradual decrease in absorption intensity. Since these stages are inseparable, we were unable to calculate any kinetic parameters (e.g., rate constants or half-life of reaction), and the approximate stability of the samples was determined by the complete disappearance of cluster-related bands at 296 nm for Mo^x^@NaOxDexQ(n) and at 320 nm for W^x^@NaOxDexQ(n) (Table 1).

One can see that, similar to pure complexes [29], W^100^@NaOxDexQ(n) samples are more stable than Mo^100^@NaOxDexQ(n). Furthermore, materials based on NaOxDex60(n) are in general 2–3 times more stable than those based on NaOxDex6(n) (Figure 4, Appendix A). The noticeable effect of oxidation degree (n) on stability was observed only in the case of less stable Mo^100^@NaOxDexQ(n) (Figure 4, Table 1), with less oxidized polymers (n = 1) providing higher stability. As was mentioned earlier, oxidation degree (n) strongly affects both functional group loading and the molecular weight of the polymer. To simplify the comparison, let us first consider a group of the polymers having similar M_w_ (~4 kDa) and different content and ratio of functional groups, namely, NaOxDex6(1), NaOxDex6(2), and NaOxDex60(2). NaOxDex6(2) contains ~1.5 times more carboxyl groups than NaOxDex6(1), while the amount of carbonyl groups is equal for both polymers (Appendix A). The lower stability of Mo^100^@NaOxDex6(2) in comparison with Mo^100^@NaOxDex6(1) (7 vs. 13 h) indicates some negative effect of acidic groups, at least when their content exceeds a certain limit. This behavior can be attributed to the low resistance of the Mo-O(O)C bond towards hydrolysis, which is also confirmed by our experience in cluster chemistry and by the absence of reports on stable water-soluble octahedral complexes with carboxylate ligands in the literature. On the other hand, NaOxDex6(1) has equal to NaOxDex60(2) loading of acidic groups along with two times lower loading of carbonyl groups (Appendix A), while the stability of Mo^100^@NaOxDex60(2) noticeably exceeds the stability of Mo^100^@NaOxDex6(1) (19 vs. 13 h). This indicates the positive impact of aldehyde groups on the stability of the materials. To reliably confirm the importance of carbonyl groups, additional experiments were conducted. Prior to the impregnation process, OxDex60(2) was treated with NaBH_3_CN to selectively reduce aldehydes. According to titration, ~40% of carbonyls were reduced during the reaction, while FTIR data confirm the preservation of acidic groups (Appendix A). The stability of Mo^100^@NaOxDex60(2)-red in culture media was assessed qualitatively by determining the time of sample precipitation. According to the observations, the sample completely precipitates after ~3 h, confirming a high contribution of aldehyde groups to the overall stability of the cluster-containing materials. From this point of view, NaOxDex60(1), having even fewer carboxylic groups, a sufficient number of carbonyls, and an increased molecular weight, should be the most promising in terms of stabilizing efficiency. Indeed, Mo^100^@NaOxDex60(1) demonstrated the highest stability among the studied samples (23 h) (Table 1). Studying the samples with x = 10 and 50 revealed a decrease in the overall stability of the materials (Table 1, Appendix A).

### 2.3. Luminescence Properties

Since the materials based on NaOxDex60(1) demonstrated the highest stability, its luminescent properties were studied only for M^x^@NaOxDex60(1) with x = 50 or 100. Therefore, luminescence spectra were recorded, and quantum yields of luminescence were determined for powdered materials as well as for fresh and aged for 8 days solutions in phosphate buffered saline (PBS) (Figure 5, Appendix A, Table 2). All samples exhibit red luminescence typical for cluster complexes with a wide profile of emission spectra. Quantum yields of luminescence of the materials in the solid state are poorly influenced by the cluster content and are in the 6–7% range for Mo and the 3–6% range for W. Usually, cluster complex incorporation into a matrix (especially when ligand substitution occurs) leads to a change in the luminescent properties, e.g., profile shape, emission maximum (λ_max_), lifetime (τ_em_), and quantum yield (Φ_em_) [31,37,38]. Here, some decrease in Φ_em_ in comparison with initial clusters is most likely due to ligand substitution, whether by water molecules or by functional groups of the oxidized dextran, which was also confirmed by FTIR. Luminescent properties in solution strongly depend on the presence of molecular oxygen, which is linked to the formation of singlet oxygen via energy transfer [39,40,41]. Quantum yields of luminescence in argon-saturated PBS for both cluster complexes are in the 2–4% range, while in aerated solution, luminescence is quenched and the quantum yields are <1%. For materials aged in PBS solution for 8 days, a noticeable emission remains (albeit only in the Ar atmosphere in the case of W-based materials), and the maximum emission is slightly shifted to the red region. Probably, such changes are associated with a change in the ligand environment of cluster complexes in solution. A similar tendency to bathochromic shift of the emission maxima during hydrolysis was previously demonstrated for pure molybdenum clusters [30] and for materials based on DNA and tungsten clusters [25].

### 2.4. Biological Properties

Despite the fact that incorporation of [{Mo_6_I_8_}(DMSO)_6_](NO_3_)_4_ in dextran matrix significantly increases the stability of the complex in aqueous solution, the stability in culture medium is still insufficient. Thus, biological studies were conducted only for the material based on the tungsten cluster complex—W^100^@NaOxDex60(1). Since the initial [{W_6_I_8_}(DMSO)_6_](NO_3_)_4_ gradually undergoes hydrolysis in aqueous media, which strongly affects its dark and photoinduced cytotoxicity [29], it was necessary to study the effect of incubation time on the biological effects of the material. Thus, the dependency of dark cytotoxicity of W^100^@NaOxDex60(1) on incubation time (0.5, 2, 5, 14, and 24 h) at a concentration range of 0.039–20 mg∙mL^−1^ was studied using an MTT assay on a model Hep-2 cancer cell line (human larynx carcinoma) (Figure 6). One can see in Figure 6 that after 0.5, 2, and 5 h of incubation, the material shows no cytotoxic effect. An increase in the incubation time up to 14 h and more leads to a slight decrease in cell viability, but IC_50_ values were not reached in the range of studied concentrations, i.e., IC_50_ is greater than 20 mg∙mL^−1^.

To compare the effect of W^100^@NaOxDex60(1) on the viability of the Hep-2 cells with the initial cluster complex [{W_6_I_8_}(DMSO)_6_](NO_3_)_4_ [29], IC_50_ values in mg∙mL^−1^ were recalculated into molar concentration of cluster cores (Table 3) using ICP-AES data. According to the results obtained, dextran significantly reduces the cytotoxicity of the complex compared to both fresh and 4-day-old solutions of [{W_6_I_8_}(DMSO)_6_](NO_3_)_4_. We believe that such effects are related to the slowing of the hydrolysis rate of the initial complex in the culture medium after coating it with dextran. Indeed, recently we showed a similar trend in [29], where a freshly prepared solution of the cluster (which contains mainly non-hydrolyzed forms) demonstrated low cytotoxicity, whereas after aging the solution for 4 days (which contains partially and fully hydrolyzed forms), the toxicity was strongly increased. Furthermore, such a great decrease in the toxicity of the material in comparison with the initial complex (>1.06 vs. >2.50 mM({W_6_I_8_} for [{W_6_I_8_}(DMSO)_6_](NO_3_)_4_ and W^100^@NaOxDex60(1) correspondingly) is most likely due to an increase in the overall solubility of the cluster after coating with dextran, i.e., the maximum concentration available for biological studies is significantly higher in the case of W^100^@NaOxDex60(1).

Cell penetration was evaluated using flow cytofluorimetry. Cells were incubated with an aqueous solution of W^100^@NaOxDex60(1) at a non-toxic concentration (5 mg mL^−1^) for 5, 14, and 24 h. According to the results obtained, after incubation for 5 h, the material penetrates into 3.5% of the cells, while after 14 h, this value increases to 30% and reaches a plateau (Figure 7). The data obtained is consistent with the cytotoxicity results, since noticeable cytotoxicity was observed only after 14 h of incubation.

Due to their efficient singlet oxygen (^1^O_2_) generation, octahedral halide cluster complexes can act as agents for photodynamic therapy [19,21,22,24,25,26,30]. Thus, we decided to evaluate the photoinduced cytotoxicity of W^100^@NaOxDex60(1) on the Hep-2 cancer cell line. The cells were incubated for 14 h (to achieve maximum uptake) with an aqueous solution of the material at a nontoxic concentration (5 mg∙mL^−1^), and then irradiated with white light for 30 min (λ = 400–800 nm, 220 mW cm^−2^). According to the data obtained, the material based on the tungsten cluster complex and oxidized dextran exhibited a noticeable phototoxic effect comparable with the effect of a 4-day-old solution of [{W_6_I_8_}(DMSO)_6_](NO_3_)_4_ but inferior to the effect of a fresh complex solution [29]—the number of viable cells decreased to 67% (Figure 8). Since biological effects of W_6_ clusters in terms of phototoxicity induced by light irradiation are barely studied in the literature, we can compare the results obtained only to other relative Mo_6_ clusters [21,22,24,26,27,33]. Overall, it is confirmed that W^100^@NaOxDex60(1) demonstrates only a moderate phototoxic effect. Two assumptions can be put forward to explain such a decrease in phototoxicity in comparison with other systems: (1) partial substitution of the ligand environment can cause a decrease in luminescent and photosensitizing properties, as was shown in a number of studies [21,31,38]; (2) a matrix can provide a shielding effect, which has a positive effect on the stability of cluster units but at the same time prevents access of molecular oxygen to the complex [31,42], which is essential for the production of ^1^O_2_. To conclude, the coverage of the cluster complex with dextran to some extent interferes with its photosensitizing properties, but significantly increases the overall stability of the system.

## 3. Materials and Methods

The drug Rheopolyglucinum (“Belmedpreparaty”, Minsk, Belarus) (saline solutions of dextran) was used as a source of dextran with molecular masses 50–70 kDa (Dex60). A solid sample of the polymer was obtained by evaporating water from the drugs at 95 °C. Dextran with a molecular mass of 6 kDa (Dex6) was purchased from Alfa Aesar (Ward Hill, MA, USA). For the sake of a fair comparison with the Dex60 sample, which contained sodium chloride after water evaporation from Rheopolyglucinum, the equivalent amount of NaCl (13.6 mg for 100 mg of polymer) was added to Dex6. [{M_6_I_8_}(DMSO)_6_](NO_3_)_4_ (M = Mo, W) were prepared according to the earlier described procedure [29]. All other reagents and solvents were purchased from Alfa Aesar, Sigma-Aldrich, or Fluka and were used as received without further purification. The concentration of hydrogen peroxide in a stock solution was 30%.

A human larynx carcinoma cell line (Hep-2) was purchased from the State Research Center of Virology and Biotechnology VECTOR and cultured in Eagle’s Minimum Essential Medium (EMEM) and Dulbecco’s Modified Eagle’s Medium (DMEM) in a ratio of 1:1 supplemented with 10% fetal bovine serum under a humidified atmosphere (5% CO_2_ and 95% air) at 37 °C.

Molybdenum, tungsten, and sulfur content in all samples was determined using inductively coupled plasma atomic emission spectroscopy (ICP-AES) using a high-resolution spectrometer, the iCAP-6500 (Thermo Scientific, Waltham, MA, USA), with a cyclone-type spray chamber and “SeaSpray” nebulizer. The spectra were obtained by axial plasma viewing. Standard operating conditions of the ICP-AES system: power = 1150 W, injector inner diameter = 3 mm, carrier argon flow = 0.7 L∙min^−1^, accessorial argon flow = 0.5 L∙min^−1^, cooling argon flow = 12 L∙min^−1^, number of parallel measurements = 3, and integration time = 5 s. Deionized water (R ≈ 18 MΩ) was used to prepare sample solutions. FTIR spectra were recorded on a Bruker Vertex 80 as KBr disks. The molecular weights (M_w_) of materials were determined by gel permeation chromatography on an Agilent-LC 1200 chromatograph equipped with a PL-aquagel-OH mixed C column and a refractive index detector. A 0.7% solution of NaN_3_ in deionized water was used as an eluent at a flow rate of 1 mL min^−1^. Calibration was performed using a polyethylene glycol/poly(ethylene oxide) calibration kit (Agilent, Santa Clara, CA, USA). Polymers were dissolved in deionized water at a concentration of 4 mg mL^−1^ and kept for 12 h in a refrigerator prior to analysis to ensure complete dissolution. Absorption spectra were recorded on an Agilent Cary 60 spectrophotometer.

Luminescence spectra and quantum yields of the materials were recorded using a Quantaurus QY C11347-1 spectrometer (Hamamatsu, Hamamatsu City, Shizuoka, Japan). The samples were measured in the solid state in air or in air-/argon-saturated PBS at a concentration of 0.1 mg mL^−1^.

### 3.1. Dextran Oxidation (OxDexQ(n))

Dextran oxidation was carried out according to the method described in [19]. Briefly, 250 mg of DexQ (where Q is the average molecular weight of the initial polymer in kDa and equals 6 and 60 kDa) was dissolved in double distilled H_2_O (2.5 mL), and hydrogen peroxide (0.65 mL or 2.6 mL) was added to the dextran solution. The final concentrations of H_2_O_2_ are 2.0 M and 5.0 M, respectively. The reaction mixture was poured into a Petri dish and kept at 100 °C until completely dry. Safety disclaimer: As heating H_2_O_2_ solution facilitates oxygen production and hence poses an explosion hazard, it is critical that the reaction be performed in an open vessel. The use of a safety shield and appropriate PPE (e.g., a safety mask) is required. The obtained samples are assigned OxDexQ(n), where n = 1 and 2, which correspond to H_2_O_2_ concentrations of 2.0 M and 5.0 M, respectively.

### 3.2. Synthesis of Oxidized Dextran Sodium Salts (NaOxDexQ(n))

The synthesis of sodium salts oxidized by dextrans was carried out according to the method described in [19]. Briefly, 100 mg of OxDexQ(n) was dissolved in H_2_O (1 mL) and treated dropwise with 0.5 M NaOH until a neutral pH was reached (pH = 7). The resulting product was precipitated by the addition of 10 mL of ethanol, washed three times with 10 mL of ethanol, and dried in the air.

### 3.3. Immobilization of [{M_6_I_8_}(DMSO)_6_](NO_3_)_4_ (M = Mo, W) into Dextrans

All experiments were carried out at room temperature. The synthesis of M^x^@DexQ and M^x^@NaOxDexQ(n) (where x = 100 for DexQ and x = 10, 50, and 100 for NaOxDexQ(n) relates to the loading of cluster complexes [{M_6_I_8_}(DMSO)_6_](NO_3_)_4_ (M = Mo or W) in milligrams per 100 mg of initial polymers) was obtained according to the following procedure: 100 mg of dextran was dissolved in 4 mL of anhydrous DMSO, and 100, 500, or 1000 µL (for x = 10, 50, or 100, respectively) of cluster complex solution in anhydrous DMSO (C = 100 mg∙mL^−1^) was added to the polymer. The total volume of the solution was adjusted to 5 mL. The resulting solution was stirred for 24 h. The reaction mixture was then slowly added to ethanol (25 mL) under intensive stirring to produce colored (orange for molybdenum and yellow for tungsten clusters) powder products. The precipitate was centrifuged (16,000 rpm for 3 min), washed with ethanol several times until it became colorless, and dried on air. For additional purification, the samples were redissolved in 1 mL of H_2_O, reprecipitated with 10 mL of ethanol, washed three times with 10 mL of ethanol, and dried in air.

### 3.4. Stability in Culture Medium

M^x^@DexQ or M^x^@NaOxDexQ(n) (10 mg) was dissolved in 50 µL of water and added to 450 µL of culture medium without dye (Dulbecco’s Modified Eagle’s Medium (DMEM)), and the resulting solutions were stored in the dark. Prior to analysis, all solutions were centrifuged (16,000 rpm for 3 min), and after that, 30 µL of the obtained solution was diluted 200 times with water. UV-vis spectra were recorded on a Cary 60 UV-vis Spectrophotometer (Agilent, Santa Clara, CA, USA) depending on time.

### 3.5. Reduction of OxDex60(2) (NaOxDex60(2)-Red) and Its Immobilization with Clusters (Mo^100^@NaOxDex60(2)-Red)

500 mg of oxidized dextran OxDex60(2) was dissolved in 5 mL of water, 500 mg of NaBH_3_CN was added, and the resulting solution was stirred for 24 h. The reduced NaOxDex60(2) (NaOxDex60(2)-red) was then precipitated with the excess of acetone (50 mL). The precipitate was separated from the solution by centrifugation (16,000 rpm for 3 min), washed with acetone three times, once with ethanol, and dried in air. During the reduction reaction, the sodium salt of oxidized dextran is formed. Immobilization with [{Mo_6_I_8_}(DMSO)_6_](NO_3_)_4_ to obtain Mo^100^@NaOxDex60(2)-red and its stability was conducted as described for DexQ and NaOxDexQ(n).

### 3.6. Biological Studies

#### 3.6.1. MTT-Assay

Hep-2 cells were seeded into 96-well plates at the concentration of 5–7 × 10^3^ cells per well and then incubated for 24 h under 5% CO_2_ atmosphere at 37 °C. The cells were treated by the aqueous solutions of W^100^@NaOxDex60(1) with resulting concentrations range of 0.04–20 mg∙mL^−1^ and incubated for 0.5, 2, 5, 14, and 24 h. The 3-(4,5-dimethylthiazol-2-yl)-2,5-diphenyltetrazolium bromide (MTT) was added to each well to achieve final concentration of 250 μg∙mL^−1^, and the plates were incubated for 4 h. The formazan formed was then dissolved in DMSO (100 μL). The optical density was measured with a plate reader Multiskan FC (Thermo Scientific, Waltham, MA, USA) at the wavelength of 570 nm. The experiment was repeated three times on separate days. The proliferation index was calculated as follows—experimental optical density (OD) value × 100/control OD value.

#### 3.6.2. Flow Cytometry (FACS) Analysis

To evaluate cellular uptake kinetics of W^100^@NaOxDex60(1), Hep-2 cells were seeded in 6-well plates at 10 × 10^4^ cells per well for each time point and were incubated for 24 h. Aqueous solution of W^100^@NaOxDex60(1) was added to cells to give a final concentration of 5 mg∙mL^−1^ and the cells were incubated for 5, 14, and 24 h at 37 °C under a 5% CO_2_ atmosphere. After treatment the cells were trypsinizated and resuspended in fresh phosphate buffered saline (PBS) with 10% FBS. Cell suspensions were analyzed using CytoFLEX (Beckman Coulter, Brea, CA, USA). A 375 nm excitation source was used with a 695 nm emission filter to visualize localization of clusters. Gating was utilized using a negative sample, and the data were expressed as median fluorescence intensity. All of the data were the mean fluorescence obtained from a population of 10,000 cells.

#### 3.6.3. Evaluation of the Photoinduced Cytotoxicity

Hep-2 cells were seeded in 96-well plates at the density of 5–7 × 10^3^ cells per well and then incubated for 24 h under 5% CO_2_ atmosphere at 37 °C. The cells were treated with the aqueous solutions of W^100^@NaOxDex60(1) at the concentrations below the dark toxicity (5 mg mL^−1^) and incubated for 14 h. After that, the cells were irradiated with a light source L8253 (Hamamatsu, Hamamatsu City, Shizuoka, Japan) (400–800 nm, 220 mW∙cm^−2^) at a distance of 20 cm for 30 min. Cells cultured in the medium without material served as a negative control.

### 3.7. Statistical Analyses

Statistical analyses were performed using the Mann-Whitney U test for unpaired data and P values of less than 0.01 were considered as significant. Data are presented as means ± SEM (standard error of the mean).

## 4. Conclusions

Thus, a series of luminescent materials based on cluster complexes [{M_6_I_8_}(DMSO)_6_](NO_3_)_4_ (M = Mo, W) and various oxidized dextrans were obtained. The incorporation of the clusters into water-soluble polymers leads to a significant stabilization of the complexes in aqueous solutions, including culture medium. The amount and ratio of functional groups strongly affect stabilization efficiency; e.g., the presence of aldehyde groups in the composition of the oxidized polysaccharide was shown to play an important role in binding to the cluster complex, while acidic groups have some negative impact on stability. All the materials demonstrate bright red/orange oxygen-sensitive emission, which is preserved even after 1 week in PBS. The most stable material, W^100^@NaOxDex60(1), was chosen for biological studies, such as dark and photoinduced cytotoxicity and cellular uptake kinetics. It was shown that covering of [{W_6_I_8_}(DMSO)_6_](NO_3_)_4_ with dextran strongly reduces its dark cytotoxicity in comparison with the initial cluster. The resulting material penetrates through the cell membrane and demonstrates a moderate photodynamic effect, which makes it relevant in the fields of bioimaging and PDT.

## Figures and Tables

**Figure 1 ijms-24-10010-f001:**
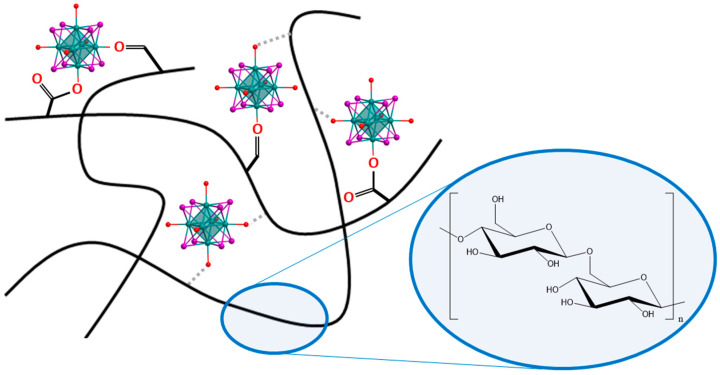
Schematic representation of possible interactions between cluster complexes and oxidized dextran.

**Figure 2 ijms-24-10010-f002:**
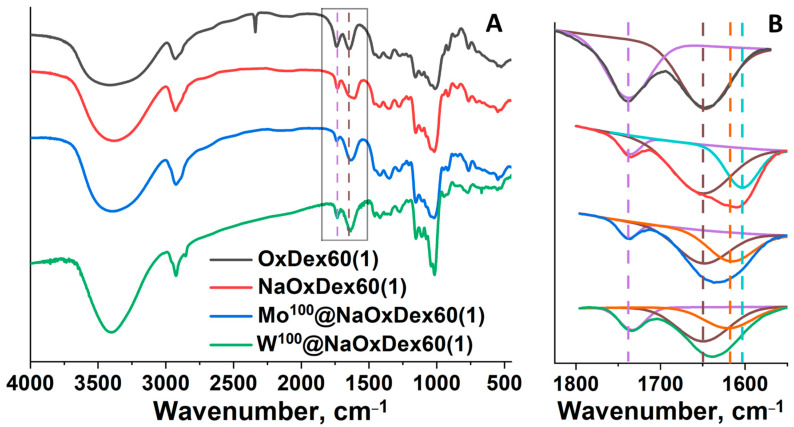
(**A**) FTIR spectra of OxDex60(1), NaOxDex60(1), and M^100^@NaOxDex60(1) (M = Mo, W). (**B**) Enlarged part of the corresponding FTIR spectra. Color code: purple—CO groups of carbonyls/free carboxylates; brown—H_2_O δ; cyan—carboxylates sodium salts; orange—coordinated carboxylates.

**Figure 3 ijms-24-10010-f003:**
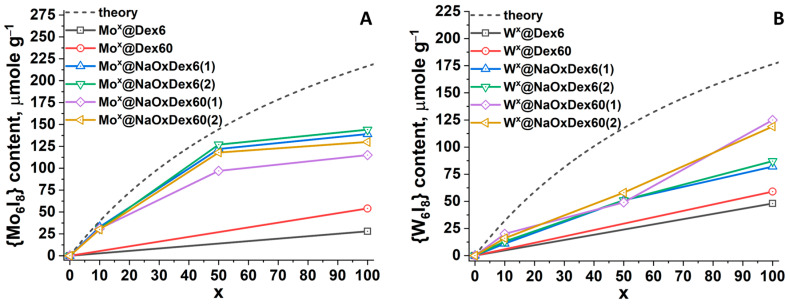
The content of {M_6_I_8_} in M^x^@DexQ and M^x^@NaOxDexQ(n); M = Mo (**A**) and W (**B**).

**Figure 4 ijms-24-10010-f004:**
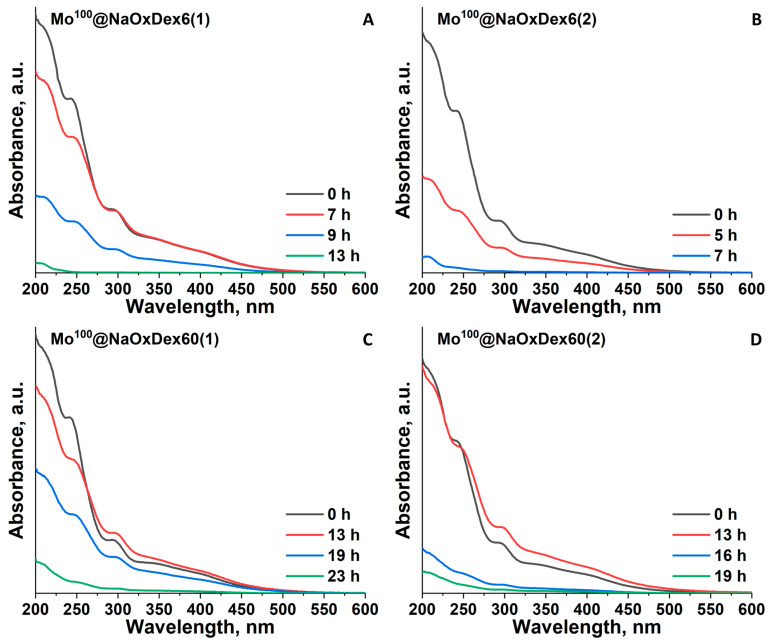
UV-vis spectra of Mo^100^@NaOxDexQ(n) ((**A**) Q = 6, n = 1; (**B**) Q = 6, n = 2; (**C**) Q = 60, n = 1; (**D**) Q = 60, n = 2) in DMEM culture medium over time.

**Figure 5 ijms-24-10010-f005:**
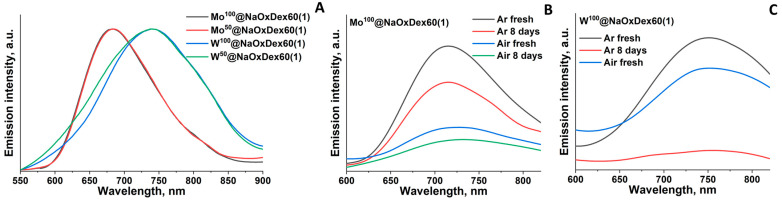
Emission spectra of solid samples M^x^@NaOxDex60(1) with x = 50 or 100 (**A**); emission spectra of fresh and 8-day-old solutions of Mo^100^@NaOxDex60(1) in PBS (in air and Ar atmosphere) (**B**); emission spectra of fresh and 8-day-old solutions of W^100^@NaOxDex60(1) in PBS (in air and Ar atmosphere) (**C**).

**Figure 6 ijms-24-10010-f006:**
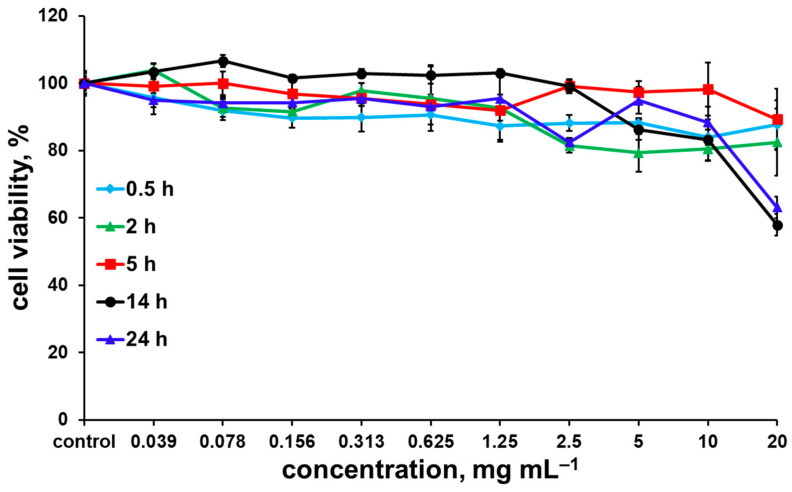
The dependency of dark cytotoxicity of W^100^@NaOxDex60(1) on incubation time against Hep-2 cells. The data represent the mean and standard deviation of three independent experiments.

**Figure 7 ijms-24-10010-f007:**
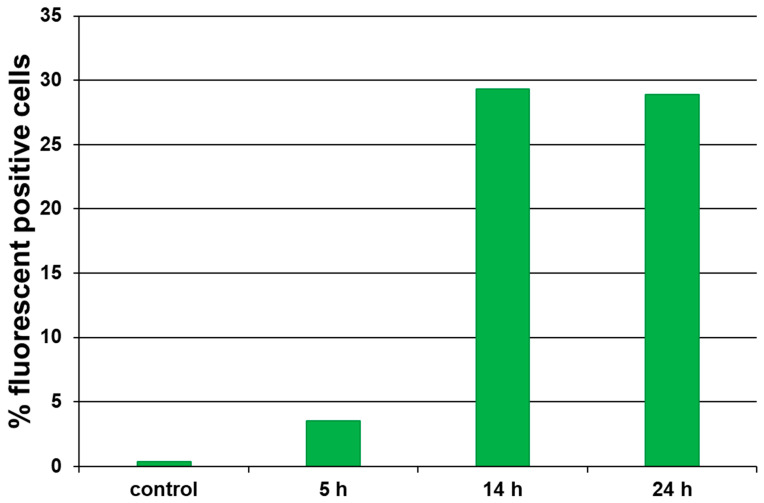
Percentage of fluorescence-positive Hep-2 cells after incubation with W^100^@NaOxDex60(1) for 5, 14, and 24 h.

**Figure 8 ijms-24-10010-f008:**
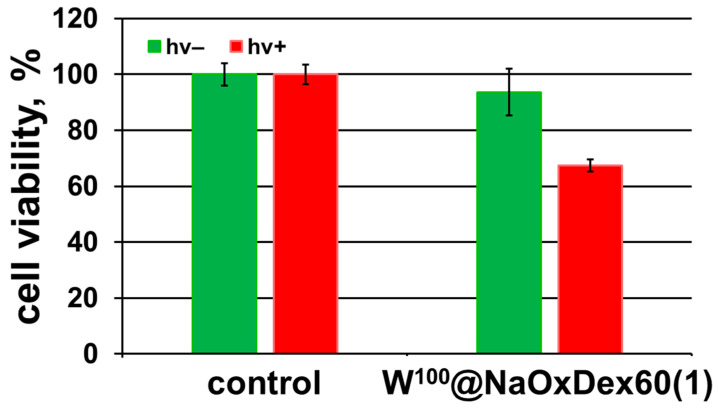
Photoinduced toxicity of W^100^@NaOxDex60(1) towards Hep-2 cells under white light irradiation (λ = 400–800 nm, 220 mW cm^−2^). Error bars represent the standard error of the mean cell viability for at least three replicate measurements.

**Table 1 ijms-24-10010-t001:** Approximate time during which materials are stable in DMEM culture medium.

Sample	Complete Disappearance of Clusters Absorption Band *, h		
	x = 10	x = 50	x = 100
Mo^x^@Dex6	-	-	unstable ^#^
Mo^x^@NaOxDex6(1)	n.d. ^$^	n.d. ^$^	~13
Mo^x^@NaOxDex6(2)	n.d. ^$^	n.d. ^$^	~7
Mo^x^@Dex60	-	-	unstable ^#^
Mo^x^@NaOxDex60(1)	~23	~23	~23
Mo^x^@NaOxDex60(2)	~13	~13	~19
W^x^@Dex6	-	-	unstable ^#^
W^x^@NaOxDex6(1)	n.d. ^$^	n.d. ^$^	~54
W^x^@NaOxDex6(2)	n.d. ^$^	n.d. ^$^	~54
W^x^@Dex60	-	-	unstable ^#^
W^x^@NaOxDex60(1)	~96 (~4 d)	~96 (~4 d)	~144 (~6 d)
W^x^@NaOxDex60(2)	~96 (~4 d)	~96 (~4 d)	~144 (~6 d)

* Absorbance was monitored at 296 nm for Mo^x^@NaOxDexQ(n) and at 320 nm for W^x^@NaOxDexQ(n); ^$^ n.d.—the experiment was not conducted; ^#^ unstable means that the sample quickly precipitates from the solution.

**Table 2 ijms-24-10010-t002:** Photophysical characteristics of M^x^@NaOxDex60(1) and [{M_6_I_8_}(DMSO)_6_](NO_3_)_4_ (M = Mo, W).

Sample	x	Solid		In PBS, Fresh			In PBS, 8 Days		
		λ_max_, nm	Φ_em_	λ_max_, nm	Φ_em_		λ_max_, nm	Φ_em_	
					Air	Ar		Air	Ar
[{Mo_6_I_8_}(DMSO)_6_](NO_3_)_4_	-	681	0.19	– ^#^	–	–	–	–	–
Mo^x^@NaOxDex60(1)	50	684	0.07	710	<0.01	0.04	715	<0.01	0.03
	100	684	0.06	715	<0.01	0.02	716	<0.01	0.02
[{W_6_I_8_}(DMSO)_6_](NO_3_)_4_	-	631	0.18	668 *	0.02 *	–	–	–	–
W^x^@NaOxDex60(1)	50	740	0.03	744	<0.01	0.03	~750 ^$^	<0.01	<0.01
	100	740	0.06	751	<0.01	0.03	~750 ^$^	<0.01	<0.01

^#^ in culture medium [30], * in water [29], ^$^ noticeable emission was observed only in the Ar atmosphere.

**Table 3 ijms-24-10010-t003:** The dependence of IC_50_ values (mM({W_6_I_8_})) of W^100^@NaOxDex60(1) and [{W_6_I_8_}(DMSO)_6_](NO_3_)_4_ (fresh and 4-day-old solutions) on incubation time.

Sample	Incubation Time, h				
	0.5	2	5	14	24
W^100^@NaOxDex60(1)	>2.50	>2.50	>2.50	>2.50	>2.50
[{W_6_I_8_}(DMSO)_6_](NO_3_)_4_ (fresh solution)	– ^$^	– ^$^	>1.06	– ^$^	>1.06
[{W_6_I_8_}(DMSO)_6_](NO_3_)_4_ (4-day-old solution)	– ^$^	– ^$^	0.94 ± 0.03 ^#^	– ^$^	0.89 ± 0.01

^$^ the experiment was not conducted; ^#^ incubated for 6h [29].

## Data Availability

The data that support the findings of this study are available from the corresponding author upon reasonable request.

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
