# Peer review of "Multifunctional Oxidized Dextran as a Matrix for Stabilization of Octahedral Molybdenum and Tungsten Iodide Clusters in Aqueous Media"

_ijms, 2023, doi:10.3390/ijms241210010_

Round 1
Reviewer 1 Report
This MS describes the utilization of a modified dextran polymer to stabilize octahedral molybdenum hexanuclear clusters (Mo6). The basis of the work is the previously developed polymer described by the same group in 2020 (ref. 18). At that moment, the methodology to produce an oxidized dextran was proposed and a series of materials reported, with OH groups replaced by CO and COOH functions mainly. Now, such material has been used to coordinate and protect labile Mo6 nanoclusters from hydrolysis, which is one of the weak points of these interesting optically active nanoobjects. The resulting hybrid material has been thoroughly characterized by a great number of techniques. The increased stability of the Mo6 entrapped in the polymer has been unequivocally demonstrated. The results here presented can be considered as a new form of stabilization for the Mo6 systems, in addition to others reported before. Moreover, the authors use the dextran-Mo6 ensemble for biological purposes, like internalization in Hep-2 cells and photoinduced toxicity on the same cells. Although the described phototoxicity is not particularly high, the materials here reported are at least promising, due to the efficient capture of Mo6. It would be nice to compare the phototoxic effect with others reported in the recent literature (specifically of Mo6 systems used for the generation of cytotoxic singlet oxygen against cancer cells). Regarding the uptake, it also would be advisable to see microscopy images of the cell cultures, if possible.
Author Response
Thank you for your positive feedback. We compared phototoxic effect with others systems containing Mo6 clusters (see 3.4 Biological properties). Also, we tried to use confocal microscopy to localize clusters inside the cells, but unfortunately their emission is strongly quenched in the cytoplasm, making it too weak to get images of acceptable quality. In turn, flow cytofluorimetry is more sensitive method providing relevant data on penetration degree.
Reviewer 2 Report
Comments and Suggestions for Authors
Therefore, the manuscript can be recommended for publication after the authors respond to my comments as listed below.
Comments
1. Title: Please correct "matrix " to "a matrix"
2. The introduction should be focused on the observations and novelty of this study, compared with other methods of preparation, and can be supported with related references. Please use this citation
https://doi.org/10.1038/s41598-021-94327-w
3. Use a unique description for the materials, and appliances used in experiments such as purity, catalog number, and manufactured name (company, city, country)
4. Please, do not use more than 3 references in one place (like [1-4], [5-10], [13-17], etc.). Either you should describe the differences.
5. Please discuss the cytotoxicity results displayed in Figure 6, section 3.4, even if just briefly. What you have done there as authors are just displayed results. There is no reason given as to why the
results come out to be what they are.
For example, let us know why you had to run the cytotoxicity test every 0.5, 2, 5, 14 and 24 h
6. Conclusions: improve this section with more appropriate information with clarity.
Minor editing of English language required
Author Response
In this manuscript “Multifunctional oxidized dextran as matrix for stabilization of octahedral molybdenum and tungsten iodide clusters in aqueous media." Overall, this is an interesting research question approach and a good dataset has been provided.
Therefore, the manuscript can be recommended for publication after the authors respond to my comments as listed below.
Answer: Thank you for your positive feedback and relevant suggestions, which we address below.
Comments
- Title: Please correct "matrix " to "a matrix"
Answer: Corrected.
- The introduction should be focused on the observations and novelty of this study, compared with other methods of preparation, and can be supported with related references. Please use this citation: https://doi.org/10.1038/s41598-021-94327-w.
Answer: Corresponding reference was added in Introduction section.
- Use a unique description for the materials, and appliances used in experiments such as purity, catalog number, and manufactured name (company, city, country).
Answer: Corrected.
- Please, do not use more than 3 references in one place (like [1-4], [5-10], [13-17], etc.). Either you should describe the differences.
Answer: According to Authors Guideline specific for IJMS, there is no limitations for the number of references used in one place. Moreover, we believe, that use of moderately large number of references demonstrates relevance of the topic and interest of scientific community to this field.
- Please discuss the cytotoxicity results displayed in Figure 6, section 3.4, even if just briefly. What you have done there as authors are just displayed results. There is no reason given as to why the results come out to be what they are. For example, let us know why you had to run the cytotoxicity test every 0.5, 2, 5, 14 and 24 h.
Answer: Corresponding section was expanded.
- Conclusions: improve this section with more appropriate information with clarity.
Answer: Conclusions section was expanded.
Round 2
Reviewer 2 Report
The authors have satisfactorily addressed most of my concerns.
The manuscript can be accepted
The manuscript can be accepted